# HIV stigma and other barriers to COVID-19 vaccine uptake among Georgian people living with HIV/AIDS: A mixed-methods study

Tamar Zurashvili [1,2], *, Tsira Chakhaia[2,3], Elizabeth J. King[4‡], Jack DeHovitz[5‡], Mamuka Djibuti[2‡]

**1** Faculty of Medicine, Ivane Javakhishvili Tbilisi State University, Tbilisi, Georgia, **2** Partnership for Research and Action for Health, Tbilisi, Georgia, **3** School of Public Health, Georgia State University, Atlanta, Georgia, United States of America, **4** School of Public Health, University of Michigan, Ann Arbor, Michigan, United States of America, **5** Department of Medicine, SUNY Downstate Health Sciences University, Brooklyn, New York, United States of America

☉ These authors contributed equally to this work.
‡ EJK, JD and MD also contributed equally to this work.
* tzurashvili@prah.ge

**Data Availability Statement:** All relevant data are within the manuscript and its Supporting Information files.

## Abstract

We conducted a study in Georgia to examine behavioral insights and barriers to COVID-19 vaccine uptake among people living with HIV (PLWH). Between December 2021-July 2022, we collected quantitative data to evaluate participants' demographics, COVID-19 knowledge, attitude, perception, and HIV stigma as potential covariates for being vaccinated against COVID-19. We conducted a multivariate analysis to define the factors independently associated with COVID-19 vaccination among PLWH. We collected qualitative data to explore individual experiences of their positive or negative choices, main barriers, HIV stigma, and preferences for receiving vaccination. Of the total 85 participants of the study, 52.9% were vaccinated; 61.2% had concerns with the disclosure of HIV status at the vaccination site. Those who believed they would have a severe form of COVID-19 were more likely to be vaccinated (OR = 23.8; 95% CI: 5.1–111.7). The association stayed significant after adjusting for sex, age, education level, living area, health care providers' unfriendly attitudes, and their fear of disclosing HIV status at vaccination places. Based on the qualitative study, status disclosure was a significant barrier to receiving care; therefore, PLWH prefer to receive COVID-19 vaccination integrated in HIV services. Conclusions: In this study, around half of the participants were not vaccinated against COVID-19. The main reasons for not being vaccinated included stigma, misleading health beliefs, and low awareness about COVID-19. An integrated service delivery model may improve vaccination uptake among PLWH in Georgia.

## Introduction

HIV/AIDS and SARS-CoV-2 (COVID-19) are two major public health problems worldwide. Compared with the HIV-negative persons, people living with HIV(PLWH) have greater risk

**Funding:** The research reported in this publication was supported by the Fogarty International Center, of National Institute of Alcohol Abuse and Alcoholism of the National Institutes of Health under Award Number D43 TW011532 to MD. The content is solely the responsibility of the authors and does not necessarily represent the official views of the National Institutes of Health. The funders had no role in study design, data collection and analysis, decision to publish, or preparation of the manuscript.

**Competing interests:** The authors have declared that no competing interests exist.

of contracting, developing complications, and dying from COVID-19 [1–3]. Therefore, vaccinating PLWH against COVID-19 is a priority public health measure to avoid complications from the disease and stop the spread of the virus. The existing literature suggests that COVID-19 vaccines are effective in PLWH [4, 5], although global uptake among this population has been reported to be 56.6% [6]. Substantial regional variations in COVID-19 vaccine uptake among PLWH are evident, with notably high rates observed in the European Region (90.1%) and the Region of the Americas (71.6%), while lower rates are reported in the Eastern Mediterranean Region (19.3%) and the African Region (35.5%) [6]. Studies also indicate that COVID-19 vaccine hesitancy (defined by the World Health Organization as the delay in acceptance or refusal of vaccination despite the availability of vaccination services [7]) is higher among this population and that factors influencing vaccine hesitancy are different for PLWH compared to HIV-negative individuals. Factors contributing to COVID-19 vaccine hesitancy among PLWH globally include perceived risks [8, 9], safety concerns [10–13], distrust in vaccine information sources, doubts about COVID-19 existence and low-risk perception [10, 14]. The study conducted among PLWH in Southwest Ethiopia indicated that the odds of intention to take the COVID-19 vaccine were 4.1 times higher among those participants who had good knowledge of COVID-19 practice compared with those who had poor knowledge [15]. The study conducted among PLWH in France reported association between a general vaccine refusal, fear of developing side effects and perception of already being immune with COVID-19 and COVID-19 vaccination hesitancy [13].

Vaccine acceptance in Eastern Europe faces challenges, with low proportions of vaccinated individuals and hesitancy influenced by factors such as public confidence in vaccine safety, efficacy, literacy, and trust in the government and medical system, underscoring the need for targeted public health initiatives to enhance COVID-19 vaccine uptake in the region [16]. In Eastern Europe and Central Asian (EECA) countries, where the HIV epidemic is escalating, our search revealed a notable gap in studies on COVID-19 vaccine acceptance, uptake and hesitancy among PLWH. This gap is particularly significant given the continued growth of the HIV epidemic in the region [17], making PLWH a crucial focus, especially considering the heightened risks associated with COVID-19. Against a backdrop of historical mistrust among citizens from the former Soviet Union towards healthcare, public institutions, medical personnel, and vaccination, a global survey identified some post-Soviet countries as having the highest hesitancy in Europe [18]. Recent research on the effects of exposure to Soviet communism on vaccine trust highlighted how socio-political regimes can adversely impact trust in vaccines, stemming from a lack of confidence in both government and healthcare providers [19]. The scarcity of data on vaccine uptake among PLWH in EECA, coupled with the unique historical context, underscores the importance of nuanced consideration in approaching vaccination attitudes. Importantly, the observed stance in the general population may not necessarily align with that of PLWH, highlighting the need for targeted research to inform tailored public health strategies for this high-risk population. The issue of COVID-19 vaccination among PLWH is important to explore in Georgia for several reasons. Foremost, overall COVID-19 vaccination rates remain suboptimal in the country. Georgia received the first batch of COVID-19 vaccine in March 2021. As of July 1st, 2022, 48.5% of adult population ages >18 was vaccinated with at least one dose and 44.9% was fully vaccinated in the same age group [20]. There is no data reported on COVID-19 vaccination specifically among PLWH in Georgia; however, we anticipate that this rate is lower compared to the general population given that little attention has been given to PLHW in the COVID-19 response in the country. To date, the country has not developed any specific plan or campaign for the vaccination of PLWH that would consider their health condition, including immune status and potential comorbidities. The other important factors impacting on vaccine uptake could be the lack of safety data on COVID-19

vaccines among PLWH as well as HIV-related stigma. To mitigate the negative effects of COVID-19 in PLWH and support the improvement of COVID-19 vaccination among this group, we explored individual level barriers by assessing the knowledge and attitude towards COVID-19 vaccination of PLWH, COVID-19, and HIV-related dual stigma and discrimination, including stigma in healthcare settings and factors associated with vaccine uptake. To explore behavioral insights and barriers to COVID-19 vaccination among PLWH, we used the Health Belief Model (HBM) [21], which has been widely used to look at COVID19 vaccine acceptance and uptake [22, 23]. The model suggests that an individual's knowledge and beliefs about health problems, perceived threats, benefits, barriers, including stigma, have a significant influence on whether and why this individual will take action to prevent or control his/her health condition.

## Methods

A cross-sectional, mixed-methods research study was conducted among adults living with HIV (>18 years old) in Georgia. The study was conducted throughout Georgia, covering the capital city, Tbilisi, as well as various regions, including Kakheti, Imereti, Guria, Samegrelo, and Ajara. Utilizing different online meeting options (e.g. Zoom, WhatsApp, Viber number, Skype, Cell Phone) during the data collection facilitated interviews with participants from diverse geographical locations.

### Quantitative phase

**Recruitment.** For the quantitative component, PLWH were recruited using a respondent-driven sampling (RDS) from the beneficiaries of a PLWH community-based organization "Real People Real Vision". Eligibility criteria for PLWH included age 18 or older with self-reported HIV positive status.

**Data collection.** Two study interviewers conducted data collection for the quantitative part of the study. Participants were offered the opportunity to take part in the interview via the different online meeting options (Zoom, WhatsApp, Viber number, Skype, Cell Phone). Once contacted via the preferred mode of communication, the study interviewer first conducted screening for eligibility using short screener questionnaire for PLWH, and after successful screening performed an interview using a structured questionnaire. To ensure effective recruitment and participation in the study, monetary incentives were provided to all participants.

**Measurements.** The structured questionnaire was developed for the quantitative data collection considering the constructs of HBM with concrete questions for each component described below.

The outcome measure for this study was self-reported COVID-19 vaccination status, dichotomized with "Yes" and "No" responses.

*Sociodemographic characteristics.* The sociodemographic characteristics were participants age, sex, area of residence, education level and financial situation.

*COVID-19 knowledge, attitude, perception.* Information on respondents' knowledge about COVID-19 was collected by asking whether COVID-19 and influenza were the same disease (respondents could respond, "Yes," "No," or "Don't know."), about transmission rout (i.e., droplets, sexual intercourse, blood borne) and existence of COVID-19 vaccines and medications ("Yes," "No," or "Don't know."). These questions had previously been used in a study on COVID-19 knowledge among Georgian population [24]. Attitude and perception questions asked about COVID-19 self-protection and avoidance ability, importance of COVID-19 vaccination, ways of improving the COVID-19 vaccination in general and among PLWH, and

about motivations for getting vaccination, self-perceived probability and severity, perceptions related to coping with stress and recovering, trust in and use of information sources. These questions were adapted from WHO guidance for conducting behavioral insights studies related to COVID-19 [25].

*HIV stigma.* For measuring HIV stigma, we used 12-item HIV Stigma Scale [26]. Subscales included personalized stigma, disclosure concerns, concerns with public attitudes and negative self-image. Personalized stigma was assessed by positive response to the following three statements: "People I care about stopped calling after learning I have HIV", "I have lost friends by telling them I have HIV" and "Some people avoid touching me once they know I have HIV". Questions on HIV status discloser asked about keeping HIV status a secret, considering it risky to disclose it and being very careful who to tell their HIV status and positive responses indicating stigmatized attitude. Concerns about public attitudes included the following statements: "most people believe a person who has HIV is dirty; "people with HIV are treated like outcasts" and "most people are uncomfortable around someone with HIV". Negative self-image due to being infected was assessed by positive response to the following three statements: "I feel guilty because I have HIV", "people's attitudes about HIV make me feel worse about myself" and "I feel I'm not as good a person as others because I have HIV". All questions had "Yes", "No", "Don't know" options for answers.

**Statistical analysis.** OpenEpi v3.03, an open-source tool, was employed to determine the sample size for this study. With a 5% margin of error and a 95% confidence interval, and considering a population size of 9,400, as estimated by the World Health Organization for people living with HIV in Georgia in 2018, the recommended sample size was determined to be 370. This calculation assumed a 50% response distribution.

The main outcome of this study was binary variable—COVID-19 vaccination status (vaccinated/not vaccinated). First, bivariate analyses were used to compare PLWH's different characteristics and outcomes. We used Chi-square and Fisher's exact tests for categorical variables and t-test and Wilcoxon rank sum for continuous variables where appropriate. Second, we used simple logistic regression with a binary variable as the outcome. For multivariable analyses, we used logistic regression to evaluate binary variable—COVID-19 vaccination status among PLWH with different characteristics. To build the model, we used stepwise regression. First, we evaluated all variables for their unique contribution in the model. Second, if one did not contribute, we removed it from the model. However, we reentered it later, if this variable was able to explain a bit more significant variance in the dependent variable than it did when it first came in. Co-variates identified by a directed acyclic graph and/or co-variates with a p-value <0.05 in the bivariate analysis were included in the multivariable analysis. We report unadjusted and adjusted odd ratios (ORs) with 95% Confidence Intervals (CIs). A p-value of <0.05 was considered significant.

## Qualitative phase

**Recruitment.** For the qualitative part we selected respondents from the pool of quantitative participants who expressed interest during the survey to subsequently participate in an in-depth interview (IDI), considering the saturation of the data.

**Data collection.** Two researchers experienced in qualitative research conducted IDIs among PLWH. A total of 20 individual IDIs were conducted between May-July 2022. PLWH participated in IDIs via different online meeting options (WhatsApp, Viber number, Skype, Cell Phone). The average duration for IDI was 1 hour. The IDIs were audio recorded with participants' consent. The recordings were then transcribed verbatim and any potential identifying information removed from the transcripts.

**Measurements.**   A semi-structured questionnaire was developed for collecting data for the qualitative phase. The topics covered by the IDI guide included the participants' individual experiences with vaccination, including probing questions about the factors that led to their positive or negative choices; main barriers, including HIV stigma and facilitators; personal attitudes toward the enhancement of the vaccination process; and preferences for the locations for receiving vaccination.

**Statistical analysis.**   The data was analyzed using an interpretive approach and categorized to identify key themes and patterns according to the HBM. Categories that followed the main topics of the interview guide, included perceived risks, attitudes barriers and facilitators that influence an individuals' decision about getting COVID-19 vaccines.

## Ethics approval

The institutional review board of the Georgian National Centre for Disease Control and Public Health (certificate IRB00002150) approved this study. All respondents were informed that their participation was voluntary and signed the informed consent form before completing the questionnaire or IDI.

## Results

### Quantitative phase

In Georgia, between December 2021—July 2022, 85 PLWH were screened for study eligibility, and all were enrolled (S1 Data). All of them completed the quantitative questionnaire. We observed a considerably low participation rates (85 enrolled vs 370 planned) in our study. This was one of the first studies among PLWH that used online interviewing, which might result in a low response rate. Another factor contributing to low participation rates could be accounted to widespread and deeply rooted stigma [27] towards HIV in Georgia, which could prevent PLWH's participation in the study (even though PLWH community-based organization supported the recruitment). Another factor for consideration is the time frame of the study, which did not allow to further continue enrolment of study participants.

The median age of the participants was 43 years (IQR = 38–48). 50.6% (n = 43) of all participants were male, and 3.5% (n = 3) did not identify themselves as a man or woman. Thirty eight percent (n = 32) and 45.9% (n = 39) of all participants had completed general secondary and high education institutions, respectively. More than half of the participants reported living in urban areas, and only a fifth of all participants lived alone. Nearly all (n = 83) claimed that their financial condition worsened or remained the same during the past three months (Table 1).

**Knowledge.**   Less than a fifth of the respondents believed that influenza and COVID-19 are the same diseases, and around 15% (n = 12) of the interviewed participants could not answer the question about the transmission route for COVID-19. Moreover, 3.5% (n = 3) of the respondents did not believe in and 14.1% (n = 12) did not know about existence of COVID-19 vaccines. Finally, only a third of all participants believed that there are no antiviral medications against COVID-19; the rest of the respondents either thought anti-COVID-19 drugs exist or were unsure about this. *Perceived Risk/Attitude*: Half of the respondents did not agree that PLWH have more probability of getting COVID-19 compared to the general public. Moreover, almost half of all respondents perceived that they would not have severe forms of disease if infected with COVID-19. Almost half of the interviewed PLWH reported feeling fear and stress. COVID-19 made 40% (n = 34) of all respondents felt total helplessness; another third of the respondents held the belief that COVID-19 is something they were not confident to combat, as they believed that avoiding COVID-19 infection to be extremely difficult. At the

**Table 1. Socio-demographic characteristics of PLWH (n = 85) and COVID-19 experience, Georgia, 2022.**

| Continuous Characteristics | Median (IQR) | Average (SD) |
|---|---|---|
| Age | 43 (38–48) | 43(9) |
| **Categorical Characteristics** | **N** | **%** |
| **Sex** | | |
| Male | 43 | 50.6% |
| Female | 39 | 45.9% |
| Other | 3 | 3.5% |
| **Education** | | |
| Incomplete general secondary education | 2 | 2.4% |
| complete general secondary education | 32 | 37.6% |
| professional education | 10 | 11.8% |
| Incomplete high education (undergraduate) | 2 | 2.4% |
| Complete high education (undergraduate) | 39 | 45.9% |
| **Living area** | | |
| Urban | 53 | 62.4% |
| Reginal Center/town | 18 | 21.2% |
| Village | 14 | 16.5% |
| **Lives alone** | | |
| Yes | 17 | 20.0% |
| No | 68 | 80.0% |
| **Financial situation over the past 3 months** | | |
| Improved | 2 | 2.4% |
| Remains the same | 28 | 32.9% |
| Worse | 55 | 64.7% |
| **Vaccinated** | | |
| Yes | 45 | 52.9% |
| No | 40 | 47.1% |
| **To your knowledge, are you, or have you been, infected with COVID-19?** | | |
| Yes | 47 | 55.3% |
| No | 38 | 44.7% |
| **Do you know people in your immediate social environment who are or have been infected with COVID-19** | | |
| Yes | 81 | 95.3% |
| No | 4 | 4.7% |
| **Do you know people someone who died from COVID-19** | | |
| Yes | 52 | 61.2% |
| No | 33 | 38.8% |

same time, almost half of the respondents perceived that the COVID-19 situation was over-hyped in the media. Along these same lines, the most trusted source of information about COVID-19 was reported to be a personal doctor among the PLHW who participated in our study. As for the attitude toward recommendations, most participants (80%, n = 68) considered mask use, hand washing, social distancing, and avoiding gatherings as activities that can help to prevent COVID-19. In contrast, less than half of the participants believed that vaccination is a way to protect oneself from COVID-19. However, almost 65% (n = 55) of participants noted that COVID-vaccination is also essential and safe for both the general public and specifically for PLWH. *HIV Stigma*: Personalized stigma was expressed by more than a quarter of the participants, with 25.9% (n = 22) stating that their close people stopped calling after learning their status, 27.1% (n = 23) lost friends and people avoid contact with them once they know

about their status. Unlike personalized stigma, HIV status disclose concerns were more prevalent in our study participants: 34.1% (n = 29) trying hard to keep their status a secret, 47.1% (n = 40) considering it risky to disclosing their status and more than 70% (n = 62) being very careful who they tell their HIV status. In terms of concerns about public attitudes 44% (n = 37) of our study participants think that most people believe a person who has HIV is dirty; 56.5% (n = 48) stated that people with HIV are treated like outcasts and 74% (n = 63) indicated that most people are uncomfortable around someone with HIV. PLWH's negative self-image due to being infected ranged from 8% (n = 7) feeling not being a good person because of HIV to 35% (n = 30) feeling worse about themselves due to people's attitudes about HIV. *Benefits and barriers*: For most participants (72.9%, n = 62), the main benefit of the COVID-19 vaccination was that after vaccination, they would not have severe forms of COVID-19. As for barriers, 61.2% (n = 52) considered the risk of disclosing their HIV status to vaccine providers as the main barrier. Supposedly due to this barrier, most of the respondents (72.9%, n = 62) thought that having vaccination services at HIV service sites would help to improve the COVID-19 vaccination among PLWHIV. It is worth noting that PLWH did not consider HIV status-related health conditions as a barrier to vaccination (Table 2).

Slightly more than half (52.9%; n = 45) of all respondents were vaccinated. In the univariate analyses, being vaccinated was significantly associated with participants' belief of the probability of developing the severe form of COVID-19 in case of getting infected and living in urban area (Table 3).

The results of the multivariate logistic regression analyses showed that, those who believed they would have a severe form of COVID-19 were more likely to be vaccinated (OR = 23.8; 95% CI: 5.1–111.7). The association stays significant and even increases after adjusting for sex, age, education level, and living area (aOR = 25.0; 95% CI: 5.0–125.0). In the third model, we included the variables about the participants' opinions on health care providers' friendly/non-friendly attitudes at vaccination places and fairness in disclosing HIV status during the vaccination. The statistically significant association between vaccination status and belief whether they develop severe form becomes more robust than in the unadjusted model (aOR = 25.7; 95% CI: 5.2–127.0) (Table 4).

## Qualitative phase

A subsample of 20 participants completed an IDI in our study between June-July 2022. Among this sample, 12 reported having been vaccinated against COVID-19. Eleven participants were male. The age range was 32–68. The descriptive results below are presented according to the main topics of the IDI guide.

**Perceived risks, beliefs and attitudes.**   During the in-depth interviews, all vaccinated participants recalled the process as a positive experience with medical personnel expressing positive attitude towards them. In the majority of cases, it was their personal decision to get vaccinated. Voluntary vaccination was based on the knowledge of its benefits: "Vaccination was my personal decision. We had a vaccination training at my office, and after that I decided to definitely get vaccinated. . .". Personal doctor's advice or celebrity's behavior was a trigger for some participants to get vaccination: "I saw some very famous people who got their shots, and I knew that everyone felt good, and I considered it necessary to get the one as well, because it would be good for my health. This helped me to make a decision about vaccination. . .".

As for unvaccinated participants, similar to the results of the quantitative component of the study, they mostly talked about the uselessness of vaccine, since they could easily recover from the disease by themselves: "I have a strong immune system. . . I won't get a severe form of COVID, so I can handle by myself. . . I don't need vaccination. . .", or already had contracted

**Table 2. Knowledge, attitude and practice, and HIV stigma experience of PLWHs (n = 85), Georgia, 2022.**

| COVID-19 Knowledge | | |
|---|---|---|
| | n | % |
| **Are coronavirus and influenza the same diseases?** | | |
| Yes | 15 | 17.6% |
| No | 52 | 61.2% |
| Do not know | 18 | 21.2% |
| **COVID-19 virus transmission route according to respondents** | | |
| Droplets | 73 | 85.9% |
| Sexual intercourse | 1 | 1.2% |
| Blood Borne | 1 | 1.2% |
| Do not know | 10 | 11.8% |
| **Does a vaccine against COVID-19 exist?** | | |
| Yes, and it is available in Georgia | 67 | 78.8% |
| Yes, and it is not available in Georgia | 3 | 3.5% |
| No | 3 | 3.5% |
| Do not know | 12 | 14.1% |
| **Do antiviral medications against COVID-19 exist?** | | |
| Yes | 16 | 18.8% |
| No | 28 | 32.9% |
| Do not know | 41 | 48.2% |
| **COVID-19 attitude: how to manage** | | |
| | n | % |
| **How can you avoid COVID-19?** | | |
| Vaccination | 41 | 48.2% |
| Mask use | 64 | 75.3% |
| Social distancing | 72 | 84.7% |
| Avoiding gathering | 76 | 89.4% |
| Hand washing | 75 | 88.2% |
| **What can help to improve the C19 vaccination among PLWH?** | | |
| Make the process simpler | 10 | 11.8% |
| Make it mandatory | 36 | 42.4% |
| Provide some monetary incentives/lottery | 29 | 34.1% |
| Allow only vaccinated people in the public spaces | 29 | 34.1% |
| Having the vaccination services at HIV service sites (integrated services | 62 | 72.9% |
| **What do you think helps (helped or would help) you as an individual to get COVID-19 vaccination** | | |
| Advice/encouragement from my personal doctor | 58 | 68.2% |
| Advice/encouragement from other HCW or social worker | 26 | 30.6% |
| Advice/encouragement from peers | 26 | 30.6% |
| Messages/information from TV/social media | 13 | 15.3% |
| **How can the COVID-19 situation be handled through government policies?** | | |
| By promoting COVID-19 vaccines | 49 | 57.6% |
| By promoting COVID-19 test | 34 | 40.0% |
| By exaggeration in restrictions | 35 | 41.2% |
| **Benefits and Barriers** | | |
| | n | % |
| **What do you think are the benefits of getting vaccinated against COVID-19** | | |
| I will not get infected | 6 | 7.1% |
| I will not spread the infection | 17 | 20.0% |

(*Continued*)

**Table 2.** (Continued)

| COVID-19 Knowledge | | |
|---|---|---|
| | **n** | **%** |
| I will not have a severe form | 62 | 72.9% |
| It will help to avoid restrictions | 49 | 57.6% |
| **What do you think are the barriers to getting vaccinated against COVID-19** | | |
| Registration is difficult | 4 | 4.7% |
| Vaccination place is far | 13 | 15.3% |
| Health care providers are not friendly at vaccination place | 38 | 44.7% |
| He/she may need disclose my HIV status and he/she may not want it | 52 | 61.2% |
| Vaccines are not available | 2 | 2.4% |
| **HIV Stigma** | | |
| | **n** | **%** |
| *Personalized stigma* | | |
| **People I care about stopped calling after learning I have HIV** | | |
| Yes | 22 | 25.9% |
| No | 51 | 60.0% |
| Do not know | 12 | 14.1% |
| **I have lost friends by telling them I have HIV** | | |
| Yes | 23 | 27.1% |
| No | 53 | 62.4% |
| Do not know | 9 | 10.5% |
| **Some people avoid touching me once they know I have HIV** | | |
| Yes | 23 | 27.1% |
| No | 48 | 56.5% |
| Do not know | 14 | 16.4% |
| *Disclosure concerns* | | |
| **I work hard to keep my HIV a secret** | | |
| Yes | 29 | 34.1% |
| No | 50 | 58.8% |
| Do not know | 6 | 7.1% |
| **Telling someone I have HIV is risky** | | |
| Yes | 40 | 47.1% |
| No | 32 | 37.6% |
| Do not know | 13 | 15.3% |
| **I am very careful who I tell that I have HIV** | | |
| Yes | 62 | 73.8% |
| No | 19 | 22.6% |
| Do not know | 3 | 3.6% |
| *Concerns about public attitudes* | | |
| **Most people believe a person who has HIV is dirty** | | |
| Yes | 37 | 43.5% |
| No | 28 | 33.0% |
| Do not know | 20 | 23.5% |
| **People with HIV are treated like outcasts** | | |
| Yes | 48 | 56.5% |
| No | 24 | 28.2% |
| Do not know | 13 | 15.3% |
| **Most people are uncomfortable around someone with HIV** | | |

(*Continued*)

**Table 2.** (Continued)

| COVID-19 Knowledge | | |
|---|---|---|
| | **n** | **%** |
| Yes | 63 | 74.1% |
| No | 13 | 15.3% |
| Do not know | 9 | 10.6% |
| *Negative self-image* | | |
| **I feel guilty because I have HIV** | | |
| Yes | 13 | 15.3% |
| No | 71 | 83.5% |
| Do not know | 1 | 1.2% |
| **People's attitudes about HIV make me feel worse about myself** | | |
| Yes | 30 | 35.3% |
| No | 43 | 50.6% |
| Do not know | 12 | 14.1% |
| **I feel I'm not as good a person as others because I have HIV** | | |
| Yes | 7 | 8.2% |
| No | 76 | 89.4% |
| Do not know | 2 | 2.4% |

COVID-19 and considered it a mild disease with no complications: "I am not vaccinated. I had COVID, it was mild, and I recovered very easily. . .". In the majority of cases, unvaccinated participants stated that nothing would change their decision regarding COVID-19 vaccination.

**Barriers to COVID-19 vaccination for PLWH.** Vaccinated PLWH mostly stated that there are no barriers to COVID-19 vaccination based on their personal experience: "The process of vaccination was really good, I would improve just nothing. . .". Both vaccinated and those who were not vaccinated respondents considered lack of information and awareness about COVID-19 vaccines as a significant barrier to COVID-19 vaccination: "The biggest barrier was probably lack of awareness, everyone was talking different things: the church, doctors, media, some of them recommended vaccination and some did no, it was really confusing. . .". Another barrier stated by the respondents from both groups was HIV stigma: "Disclosing HIV status is also a problem of our society, as a rule PLWH hide information about it, and they should not, no one hides information about their diabetes, and this should be the case for HIV as well. But nobody is working on this issue, neither doctors, nor the media and it's very bad. . .", however those who were vaccinated either did not have problem with disclosure or just simply did not talk about their status to the medical personnel at the vaccination site. Majority of unvaccinated respondents stated that they simply did not want to be vaccinated, although in case they would decide to get a shot, status disclose would be a challenge for them: "I didn't even think for a second that I should get vaccinated and if I would want, that factor [status disclosure] would be a barrier for me, I live in a village, and everyone knows me here. Stigma still exists here. . ."

**Integration of services and a trustworthy person.** The majority of respondents, both vaccinated and unvaccinated, noted that integrating vaccination services in HIV treatment and care services would be convenient for PLWH: "Yes, it [integration] would be very convenient, I would feel more comfortable with my personal doctor, because I trust her. . .". However, those who already got vaccination, do not consider vaccination sites outside HIV services being an issue: "I got vaccination outside HIV care and didn't have any problem either, but I

**Table 3. Different characteristics among vaccinated and not vaccinated PLWH (n = 85), Georgia, 2022.**

| | PLWH Vaccinated | PLWH Not Vaccinated | P-value |
|---|---|---|---|
| | N (%) | N (%) | |
| **Sex** | | | 0.171 |
| Male | 24 (53.3) | 19 (47.5) | |
| Female | 18 (40.0) | 21 (52.5) | |
| Other | 3 (6.7) | 0 (0.0) | |
| **Education** | | | 0.152 |
| Incomplete general secondary education | 1 (2.2) | 1 (2.5) | |
| complete general secondary education | 17 (37.8) | 15 (37.5) | |
| professional education | 2 (4.4) | 8 (20.0) | |
| Incomplete High education | 2 (4.4) | 0 (0.0) | |
| Complete high education | 23 (51.1) | 16 (40.0) | |
| **Living area** | | | **0.0348** |
| Urban | 31 (68.9) | 22 (55.0) | |
| Regional Center/town | 11 (24.4) | 7 (17.5) | |
| Village | 3 (6.7) | 11 (27.5) | |
| **Lives alone** | | | 0.2772 |
| Yes | 11 (24.4) | 6 (15.0) | |
| No | 34 (75.6) | 34 (85.0) | |
| **Financial situation over the past 3 months** | | | 0.0561 |
| Improved | 1 (2.2) | 1 (2.5) | |
| Remains the same | 20 (44.4) | 8 (20.0) | |
| Worse | 24 (53.3) | 31 (77.5) | |
| **I will not have a severe form** | | | $< .0001$ |
| Yes | 43 (95.6) | 19 (47.5) | |
| No | 2 (4.4) | 21 (52.5) | |
| **Health care providers are not friendly at vaccination place** | | | 0.8486 |
| Yes | 19 (42.2) | 19 (47.5) | |
| No | 26 (57.8) | 21 (52.5) | |
| **He/she may need disclose my HIV status and he/she may not want it** | | | 0.2594 |
| Yes | 13 (32.5) | 27 (67.5) | |
| No | 20 (44.4) | 13 (32.5) | |
| *Personalized stigma*—**I have lost friends by telling them I have HIV** | | | 0.5172 |
| Yes | 10 (22.2%) | 13 (32.5%) | |
| No | 29 (64.4%) | 24 (60.0%) | |
| Do not know | 6 (13.3%) | 3 (7.5%) | |
| *Disclosure concerns*—**I am very careful who I tell that I have HIV** | | | 0.1356 |
| Yes | 32 (72.7%) | 30 (75.0%) | |
| No | 12 (27.3%) | 7 (17.5%) | |
| Do not know | 0 (0.0%) | 3 (7.5%) | |
| *Concerns about public attitudes*—**Most people are uncomfortable around someone with HIV** | | | 0.1744 |
| Yes | 31 (68.9%) | 32 (80.0%) | |
| No | 10 (22.2%) | 3 (7.5%) | |
| Do not know | 4 (8.9%) | 5 (12.5%) | |
| *Negative self-image*—**People's attitudes about HIV make me feel worse about myself** | | | 0.6755 |
| Yes | 17 (37.8%) | 13 (32.5%) | |
| No | 23 (51.1%) | 20 (50.0%) | |
| Do not know | 5 (11.1%) | 7 (17.5%) | |

**Table 4. Unadjusted and adjusted associations between having vaccination and health belief of not having COVID-19 severe form among PLWH (n = 85), Georgia, 2022.**

| | PLWH Vaccinated N (%) | PLWH Not Vaccinated N (%) | OR[a] (95% CI) | aOR[b] (95% CI) | aOR[c] (95% CI) |
|---|---|---|---|---|---|
| **I will not have a severe form** | | | | | |
| Yes | 43 (95.6) | 19 (47.5) | 23.8 (5.1–111.7) | 25.0 (5.0–125.0) | 25.7 (5.2–127.0) |
| No | 2 (4.4) | 21 (52.5) | Ref | Ref | Ref |

[a]OR—unadjusted Odds Ratio

[b]aOR—adjusted Odds Ratio controlled for sex, age, education level, living area

[c]aOR—adjusted Odds Ratio controlled for the variables about the participants' opinion on Health care providers' friendly/non friendly attitude at vaccination place and their fair to disclose the HIV status during vaccination process.

think it [integration] would be better, my doctor knows my status and I would be more confident. . ." While talking about most trustworthy person on getting general medical or COVID-19 and vaccination information/advice, the vast majority of participants talked only about their personal doctors/infectious disease specialist.

## Discussion

The main finding of our study was the low coverage of vaccination (only slightly more than half) among PLWH. Our finding is slightly lower than a global COVID-19 vaccine uptake among PLWH of 56.6%, but at the same time notably lower than the rate of 90.1% for the European Region as reported by Sulaiman et al. [6]. When considering the broader context, our findings also underscore a significant gap compared to the global vaccine uptake among the general population, which stands at a higher rate of 70.6% [28]. Results from our research highlight three main reasons for PLWH not being vaccinated: misleading health beliefs, low awareness about COVID-19, and stigma.

Most participants believed they have a low probability of developing severe disease after being infected with COVID-19. At the same time, for these respondents, the main benefit of the COVID-19 vaccination was that after vaccination, they would not have severe forms of COVID-19. These contradictory beliefs might cause reluctance regarding the necessity of getting COVID-19 vaccination. Unadjusted and adjusted statistically significant associations between being vaccinated and participants' opinion of the probability of developing severe COVID-19 in case of getting infected proved the above statement. In-depth interviews, during which unvaccinated respondents mostly talked about the uselessness of COVID-19 vaccines since they could quickly recover from the mild disease with no complications, also confirmed these findings. A systematic review and meta-analysis conducted by Sulaiman et al. reported association of perceived high susceptibility to SARS-CoV-2 infection and a strong belief in the vaccine's effectiveness and utility with higher COVID-19 vaccine acceptance and uptake among PLWH [6]. HBM constructs, such as perceived severity of COVID-19 infection and perceived benefits of a COVID-19 vaccine have also been linked to vaccination acceptability among general population [29, 30]. The results highlight the interconnected role of risk perception and perceived vaccine benefits in influencing vaccination behavior.

Stigma was another main reason for vaccine non-uptake among PLWH. According to our quantitative analyses, the risk of disclosing their HIV status to vaccine providers is the main barrier for HIV-positive respondents. This finding is in line with our qualitative findings. During the IDIs, respondents noted that PLWH prefer not to disclose information about their HIV status. Supposedly due to this barrier, most PLWH expressed that having vaccination services at HIV service sites would be convenient for getting a vaccination. The study conducted

among Chinese PLWH also revealed apprehensions regarding the disclosure of HIV status in the context of COVID-19 vaccination [8]. The low vaccine uptake among marginalized populations, including PLWH in India, has been attributed to factors such as HIV-related stigma, fear of disclosing HIV status, and a scarcity of communication materials tailored for these groups, exacerbating the challenge of combating COVID-19 in these vulnerable communities [31].

Finally, our study's quantitative and qualitative data showed that PLWH lack knowledge regarding COVID-19. These findings are in line with the study conducted among PLWH in Southwest Ethiopia, which revealed the importance of knowledge in acceptance of COVID-19 vaccination [15]. During the in-depth interviews, unvaccinated respondents named misinformation and lack of trusted sources as significant barriers. Their main concern was that various sources, such as media, health institutions, churches, etc., spread different recommendations regarding the COVID-19 vaccination. The findings from a study conducted in in British Columbia, Canada is also illustrative of inconsistency between the strong recommendations for COVID-19 vaccination for PLWH and existing intentions [32]. Qualitative and quantitative analyses showed that personal doctors are the most trusted source of information for PLWH. Therefore, medical doctors working with HIV-positive people should conduct more COVID-19-related education activities among their patients. The findings from a study among PLWH at a primary care clinic in an urban area of New York City highlight the significance of physicians' active engagement in individualized education with PLWH leading to an increased vaccine acceptance [33]. HCWs should mainly focus on COVID-19 and HIV dual stigma-related issues and misleading beliefs about developing severe disease among PLWH. A systematic approach to the education sessions would help to increase knowledge and raise awareness about COVID-19 among PLWH.

This study had several limitations. First, the quantitative part was a cross-sectional study, so no causality could be established. The sampling method used (non-probability) for this study makes it difficult to infer the results to the entire PLWH population in Georgia. In our study, the vaccination status was self-reported, however it has been shown that in large observational studies where medical records are not available, self-reported dates and product details for COVID-19 vaccination can be a good substitute [34].

## Conclusion

This study was the first in the ECCA region to explore the factors associated with COVID-19 vaccine uptake specifically among PLWH. According to our findings, around half of the PLWH were not vaccinated against COVID-19. We identified that the main reasons for not being vaccinated among PLWH were (1) stigma, (2) misleading health beliefs and (3) low awareness about COVID-19.

COVID-19 awareness raising campaigns and educational sessions should be continually conducted among PLWH. Medical doctors and mainly, HIV service providers as the most trusted source of information for PLWH, should work closely with their patients to support the improvement of COVID-19 vaccination among PLWH in Georgia.

## Supporting information

**S1 Data. Study dataset.** This dataset comprises anonymized records of participants involved in the study in the format of Excel spreadsheet. It includes information on demographics, COVID-19 vaccination status, knowledge, attitudes, and perceptions related to COVID-19, as well as barriers and facilitators to COVID-19 vaccination. The dataset encompasses data on

HIV and HIV and COVID dual stigma among participants.
(XLSX)

## Acknowledgments

The authors thank local PLWH community-based organization "Real People Real Vision" for assistance in the field work and recruitment of participants for this study.

## Author Contributions

**Conceptualization:** Elizabeth J. King, Jack DeHovitz, Mamuka Djibuti.

**Formal analysis:** Tsira Chakhaia.

**Investigation:** Tamar Zurashvili, Tsira Chakhaia.

**Methodology:** Tamar Zurashvili, Tsira Chakhaia, Mamuka Djibuti.

**Project administration:** Tamar Zurashvili.

**Resources:** Tamar Zurashvili.

**Supervision:** Tamar Zurashvili.

**Validation:** Tamar Zurashvili, Tsira Chakhaia, Mamuka Djibuti.

**Visualization:** Tamar Zurashvili, Tsira Chakhaia.

**Writing – original draft:** Tamar Zurashvili, Tsira Chakhaia.

**Writing – review & editing:** Tamar Zurashvili, Tsira Chakhaia, Elizabeth J. King, Jack DeHovitz, Mamuka Djibuti.

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
