## [Decision Letter · Decision Letter 0]

22 Dec 2023

PGPH-D-23-01963

HIV stigma and other factors contributing to COVID-19 vaccine hesitancy among Georgian people living with HIV/AIDS: a mixed-methods study

Dear Dr. Zurashvili,

Thank you for submitting your manuscript to PLOS Global Public Health. After careful consideration, we feel that it has merit but does not fully meet PLOS Global Public Health’s publication criteria as it currently stands. Therefore, we invite you to submit a revised version of the manuscript that addresses the points raised during the review process.

The manuscript has been evaluated by three reviewers, and their comments are available below.

The reviewers have raised a number of concerns that need attention, including requests for clarification and additional detail.

Could you please revise the manuscript to carefully address the concerns raised?

One or more reviewers has recommended that you cite specific previously published works. As always, we recommend that you please review and evaluate the requested works to determine whether they are relevant and should be cited. It is not a requirement to cite these works. 

We look forward to receiving your revised manuscript.

Kind regards,

Steve Zimmerman, PhD

PLOS Staff Editor

Journal Requirements:

Additional Editor Comments (if provided):

Reviewers' comments:

Reviewer's Responses to Questions

**Comments to the Author**

1. Does this manuscript meet PLOS Global Public Health’s publication criteria? Is the manuscript technically sound, and do the data support the conclusions? The manuscript must describe methodologically and ethically rigorous research with conclusions that are appropriately drawn based on the data presented.

Reviewer #1: Yes

Reviewer #2: Yes

Reviewer #3: Yes

2. Has the statistical analysis been performed appropriately and rigorously?

Reviewer #1: Yes

Reviewer #2: Yes

Reviewer #3: No

3. Have the authors made all data underlying the findings in their manuscript fully available (please refer to the Data Availability Statement at the start of the manuscript PDF file)?

Reviewer #1: Yes

Reviewer #2: Yes

Reviewer #3: No

4. Is the manuscript presented in an intelligible fashion and written in standard English?

Reviewer #1: Yes

Reviewer #2: Yes

Reviewer #3: No

5. Review Comments to the Author

Reviewer #1: Dear authors,

You have put tremendous work into preparing this manuscript on a topical issue. I have highlighted some issues that will improve the manuscript's acceptability in the scientific community.

1. Line 26: “Between December-July 2022” State the year of commencement i.e December 2020, December 2021 e.t.c

2. Line 46: Remove “and death”

3. Line 77: Change “not” to “no”

4. Line 143: Please, mention the specific type of regression used i.e binary, multinomial e.t.c

5. A brief description of the study area should also be made.

6. Sample size estimations should be presented. Considering inferential statistics were conducted this is mandatory

Reviewer #2: I appreciate the efforts of the authors in carrying out such a study that explores a critical area of public health and seeks to address the gap in the literature in Geargia. The mixed method approach has provided more insights into the concept and the use of Health Belief Model is appropriate. However, I have some comments that I would like the authors to address:

1. Line 68, are the authors referring to “being immune to COVID-19” OR “…being immune with COVID-19 vaccine hesitancy”

2. It is good that the authors have provided information on the lack of data on vaccine hesitancy among PLHIV in EECA region. However, the flow of the text on P3-4, L64-73 seems to be disjointed. It could be improved by rephrasing and restructuring the sentences. The authors may consider integrating those sentences into one paragraph. Also, it is worthy to note that the attitude observed in the general population may not be reflected among PLHIV.

3. P4, L74 consider changing “There are not data” to “there is no data”

4. The authors highlighted that the COVID-19 vaccine uptake rates in Georgia was suboptimal. One will wonder whether this is associated with availability and accessibility of the vaccine in the country. And to make a fairer comparison with other nations one would like to know when the rollout was commenced in the country. The authors should also consider obtaining an updated vaccination uptake rates in the country, as the data might have changed between July 2022 to date.

5. The sentence that started with “The other important factors….” in L81-83 is not well fitted in that position. According to my understanding, the paragraph was intended to provide justification for conducting the study. Can this sentence be moved to where they discussed the factors associated with vaccine acceptance/hesitancy in other parts of the introduction? This will improve the flow and coherence of the text.

6. Considering the study design, it will be more transparent and appropriate if the authors can provide justification for their sample size.

7. The number of participants given in the caption of table 1 (L184) does not correspond to the total number presented on the table. Similarly, it seems the percentage mentioned in L188 in not correct. Additionally, the authors may consider introducing subheadings in the results section to improve the readability.

8. One will wonder why the authors omitted many important sociodemographic characteristics (such as ethnicity/race, employment status, income etc.) and factors related to the HIV itself (utilization of ART, virologic suppression, etc.) and the COVID-19 vaccine (in the questionnaire), which have been proven to be essential in influencing vaccine acceptance and hesitancy. These would have provided more detailed information about the study population. In addition, have the authors attempted to verify the information provided by the participants with their medical record, especially those related to the vaccination? It is also relevant to include the number doses received by the vaccinated group.

9. The discussion section of the manuscript needs to be improved. I suggest comparing the vaccination rate obtained in this study with what was reported in other parts of the globe among the same population. Also, comparison should be made with the latest data among the general population in the same country. Moreover, since stigma is key aspect of the study, it is important to further support this finding with the available evidence (if any) on the role of stigma on vaccine acceptance/hesitancy.

Reviewer #3: I thank the Editor for giving the opportunity to review this important manuscript. The authors have made a commendable effort to fill in an important gap by reporting COVID-19 vaccination (not hesitancy0 determinants in Georgia. The findings in the study are needed to inform public health in the country. Despite making a lot of effort, I still believe the work needs a significant improvement. My comments to the authors are:

1. Lines 65-66 the authors, without supporting with a reference wrote: “Data for EECA indicates that HIV epidemic continues 66 to grow in the region”

2. In Table 1 title, the authors wrote “…..characteristics of PLWH (n=84) and COVID-19 experience”. Is the n = 84 or 85

3. In Table 2, the authors wrote “What do you think how can be handled the COVID-19 situation”. Please rephrase for clarity.

4. What approach to logistic regression modeling have the authors used? Forward, backward, or stepwise?.

5. Why have the authors not provided the full results of the bivariate and multivariate regression analysis. Please provide this for transparency even if no significant association is found it is still important.

6. The authors have provided data on important behavioural determinants of vaccination in PLHIV but have not included most of these in their regression analysis. Examples include: I will not have a severe form; Health care providers are not friendly at vaccination place; He/she may need disclose my HIV status and he/she may not want it; I have lost friends by telling them I have HIV; I am very careful who I tell that I have HIV; and overall COVID-19 knowledge and attitude. Please report the results of all analysis in full even as a supplementary information.

7. Also 45 out of the 85 participants have been vaccinated according to the authors, while 40 were unvaccinated. It should be noted that these 40 unvaccinated participants can not be said to be hesitant (unless you’ve asked them and they all said they didn’t intend to vaccinate, which you’ve not stated anywhere in your manuscript) according to the WHO SAGE Working group of experts on vaccine hesitancy (delay in acceptance or 51 refusal of vaccination despite the availability of vaccination services) stated and cited by the authors in the introduction. I am saying this because the word hesitancy is included in your manuscript title although not anywhere in the results section. Therefore, the authors need to rephrase the article title and remove the word “hesitancy” from it because its not what they evaluated. Similarly, they should apply this same change to all relevant sections of the manuscript especially the discussion section.

8. Despite only evaluating the determinants of COVID-19 vaccine uptake, the authors kept citing studies on determinants of hesitancy and acceptance in their study. The authors may wish to consider this important manuscript for guide: Sulaiman, S. K., Musa, M. S., Tsiga-Ahmed, F. I. I., Sulaiman, A. K., & Bako, A. T. (2023). A systematic review and meta-analysis of the global prevalence and determinants of COVID-19 vaccine acceptance and uptake in people living with HIV. Nature Human Behaviour, 1-15. https://doi.org/10.1038/s41562-023-01733-3.

I look forward to reading the revised version of this manuscript.

6. PLOS authors have the option to publish the peer review history of their article (what does this mean?). If published, this will include your full peer review and any attached files.

**Do you want your identity to be public for this peer review?** For information about this choice, including consent withdrawal, please see our Privacy Policy.

Reviewer #1: No

Reviewer #2: **Yes: **Muhmmad Sale Musa

Reviewer #3: **Yes: **Sahabi Kabir Sulaiman

---

## [Decision Letter · Decision Letter 1]

11 Mar 2024

HIV stigma and other barriers to COVID-19 vaccine uptake among Georgian people living with HIV/AIDS: a mixed-methods study

PGPH-D-23-01963R1

Dear Dr. Zurashvili,

We are pleased to inform you that your manuscript 'HIV stigma and other barriers to COVID-19 vaccine uptake among Georgian people living with HIV/AIDS: a mixed-methods study' has been provisionally accepted for publication in PLOS Global Public Health.

Best regards,

Muhammad Sale Musa, MBBS

Guest Editor

Dear Dr. Zurashvili,

I appreciate your patience during the review process. I also thank you for taking the time to respond to all concerns raised by the reviewers.

While your article has been considered fit for publication, I would like you to address some few issues that will potentially improve the the manuscript. These include:

1. In the introduction section, it will more appropriate to use indefinite articles to introduce some of the sentences. For example, "A study conducted among PLWH in...." sound less definitive and more appropriate in this context than "the study...."

2. From L72-87: Kindly consider replacing "underscoring" or "underscores" in some of the places with another word (synonym) to improve the readability and make the text more engaging

3. Consider correcting L175: "..with a binary variable as the outcome" instead of "with binary variable an outcome".

4. L266: Unvaccinated respondents OR "those who were not vaccinated" sound more appropriate than "Not vaccinated respondents"

5. L392 could be improved by rephrasing it to something like "This study was the first in the ECCA region to explore factors...."

Congratulations to all the authors.

Kindest regards,

Dr. Musa, Guest Editor

Reviewer Comments (if any, and for reference):

The authors have adequately addressed the previous comments.

Reviewer's Responses to Questions

**Comments to the Author**

Reviewer #3: All comments have been addressed

2. Does this manuscript meet PLOS Global Public Health’s publication criteria? Is the manuscript technically sound, and do the data support the conclusions? The manuscript must describe methodologically and ethically rigorous research with conclusions that are appropriately drawn based on the data presented.

Reviewer #3: Yes

3. Has the statistical analysis been performed appropriately and rigorously?

Reviewer #3: Yes

4. Have the authors made all data underlying the findings in their manuscript fully available (please refer to the Data Availability Statement at the start of the manuscript PDF file)?

Reviewer #3: No

5. Is the manuscript presented in an intelligible fashion and written in standard English?

Reviewer #3: Yes

6. Review Comments to the Author

Reviewer #3: The authors have remarkably improved the quality of their work. I have no further comments.

Congratulations.

7. PLOS authors have the option to publish the peer review history of their article (what does this mean?). If published, this will include your full peer review and any attached files.

**Do you want your identity to be public for this peer review?** For information about this choice, including consent withdrawal, please see our Privacy Policy.

Reviewer #3: **Yes: **Dr Sahabi Kabir Sulaiman
